# Quantification of gait parameters in freely walking wild type and sensory deprived *Drosophila melanogaster*

**César S Mendes[1], Imre Bartos[2], Turgay Akay[3], Szabolcs Márka[2], Richard S Mann[1]\***

[1]Department of Biochemistry and Molecular Biophysics; [2]Department of Physics; [3]Department of Neurological Surgery, Columbia University, New York, USA

**Abstract** Coordinated walking in vertebrates and multi-legged invertebrates such as *Drosophila melanogaster* requires a complex neural network coupled to sensory feedback. An understanding of this network will benefit from systems such as *Drosophila* that have the ability to genetically manipulate neural activities. However, the fly's small size makes it challenging to analyze walking in this system. In order to overcome this limitation, we developed an optical method coupled with high-speed imaging that allows the tracking and quantification of gait parameters in freely walking flies with high temporal and spatial resolution. Using this method, we present a comprehensive description of many locomotion parameters, such as gait, tarsal positioning, and intersegmental and left-right coordination for wild type fruit flies. Surprisingly, we find that inactivation of sensory neurons in the fly's legs, to block proprioceptive feedback, led to deficient step precision, but interleg coordination and the ability to execute a tripod gait were unaffected.

## Introduction

Ever since life became terrestrial ~360 million years ago, animals have developed increasingly sophisticated methods to navigate their environments (*Dickinson et al., 2000*). Locomotion is essential for animals to escape from predators, find mates, and search for food. One of the most common forms of terrestrial locomotion depends on the movement of multi-jointed legs. For this to occur, animal nervous systems face two main computational challenges. First, multiple leg joints must move rhythmically and in a precisely coordinated fashion to allow the stereotyped movements that occurs during the swing and stance phases of each step cycle. Second, these movements must be coordinated between legs, which number four in a typical tetrapod and six in a hexapod. Both challenges are met in part by interactions between central pattern generators (CPGs), neural networks within the central nervous system that have the capacity to generate rhythmic outputs (*MacKay-Lyons, 2002*). When used for walking, individual CPGs result in the rhythmic and alternating activity of motor neurons that control the flexion and extension of single leg joints (*Bässler, 1977*; *Strauss, 2002*; *Borgmann et al., 2009*; *Büschges et al., 2011*). Leg movement coordination is also assisted by proprioceptive sensory inputs that report the load and position of leg joints (*Bässler, 1977*; *Borgmann et al., 2009*). Other sensory modalities, such as visual, olfactory and gravitational, also modulate the activity of locomotor CPGs to allow animals to readily change their motor behavior in response to their environment (*Frye, 2010*).

An understanding of locomotion requires the identification of the neurons that comprise locomotor neural circuits. Given the complexity of these circuits, insects provide an attractive model to achieve this goal due to their relative simplicity, approachable physiology and availability of genetic tools. Moreover, adult insects share with vertebrates the same general principles of locomotion (*Pearson, 1993*). Nevertheless, despite many anatomical similarities, it is less clear how similar the neural circuitries underlying locomotion are in vertebrates and invertebrates. Groundbreaking studies in cockroaches, locusts and stick insects have identified many components and fundamental rules that regulate the

**\*For correspondence:** rsm10@columbia.edu

**Competing interests:** The authors have declared that no competing interests exist

**eLife digest** Most animals need to be able to move to survive. Animals without limbs, such as snakes, move by generating by wave-like contractions along their bodies, whereas limbed animals, such as vertebrates and arthropods, walk by coordinating the movements of multi-jointed arms and legs. Locomotion in limbed animals involves bending each joint within each arm or leg in a coordinated manner, while also ensuring that the movements of all the limbs are coordinated with each other. In bipeds such as humans, for example, it is critical that one leg is in the stance phase when the other leg is in the swing phase. The rules that govern the coordination of limbs also depend on the gait, so the rules for walking are not the same as the rules for running.

The nervous systems of bipeds and other animals that walk solve these problems by using complex neural circuits that coordinate the firing of the relevant motor neurons. Two general mechanisms are used to coordinate the firing of motor neurons. In one mechanism, local interneurons within the central nervous system coordinate motor neuron activities: in vertebrates these interneurons are found in the spinal cord. A second mechanism, termed proprioception, relies on sensory neurons that report the load and joint angles from the arms and legs back to the central nervous system, and thereby influence the firing of the motor neurons. Remarkably, both of these mechanisms, and also the types of neurons that comprise motor neuron circuits, are conserved from arthropods to vertebrates.

Mendes et al. describe a new approach that can be used to analyze how the fruit fly, *D. melanogaster*, walks on surfaces. They use a combination of an optical touch sensor and high-speed video imaging to follow the body of the fly as it walks, and also to record when and where it places each of its six feet on the surface as it moves. Then, using a software package called FlyWalker, they are able to extract a large of number of parameters that can be used to describe locomotion in adult fruit flies with high temporal and spatial resolution. Many of these parameters have never been measured or studied before.

Mendes et al. show that fruit flies do not display the abrupt transitions in gait that are typically observed in vertebrates. However, they do modify their neural circuits depending on their speed: indeed it appears that flies use subtly different neural circuitry for walking at slow, medium and fast speeds. Moreover, when genetic methods are used to block sensory feedback, the fly is still able to walk, albeit with reduced coordination and precision. Further, the data suggest that proprioception is less important when flies walk faster compared to when they walk more slowly. The next step in this research will be to combine this new method for analyzing locomotion in flies with the wide range of genetic tools that are available for the study of *Drosophila*: this will allow researchers to explore in greater detail the components of the motor neuron circuitry and their role in coordinated walking.

walking apparatus, such as the definition of multiple gaits (***Graham et al., 1985***; ***Burrows, 1992***; ***Zill et al., 2004***; ***Ritzmann and Büschges, 2007***; ***Büschges et al., 2008***). However, unlike in vertebrates, where gait transitions are discontinuous, it is less clear in which situations insects use distinct gaits and how they transition between them. Although theoretical considerations suggested the possibility that gait transitions may be more gradual than they are in vertebrates, this question has not been fully resolved (***Graham et al., 1985***). Walking speeds in cockroaches, for example, cluster into two groups, but both rely on the tripod gait (***Bender et al., 2011***). The stick insect tends to use a tetrapod gait at slower speeds and a tripod gait at faster speeds (***Graham et al., 1985***). *Drosophila*, on the other hand, are reported to primarily use the tripod gait (***Strauss and Heisenberg, 1990***). Similar questions arise with respect to the role of sensory feedback in coordinated walking. Although feedback is thought to be critical for coordination between legs (***Bässler, 1977***; ***Borgmann et al., 2009***) most studies address this question using electrophysiology readouts with tethered animals. In addition, with the exception of surgical ablation experiments (***Usherwood et al., 1968***; ***Cruse et al., 1984***), it has been difficult to dissect the contribution of individual sensory modalities. Gaining further insights into these questions may benefit by the use of systems that use freely walking animals with access to genetic tools to identify and manipulate the individual components of locomotor circuits.

The fruit fly, *D. melanogaster*, is a powerful genetic model with a large collection of mutants and an increasingly sophisticated genetic toolkit (*Pfeiffer et al., 2008*; *Venken et al., 2011*). However, despite the availability of powerful genetic tools, there is a lack of quantitative and robust methods to analyze the consequences of manipulating locomotor circuits in the fruit fly. Consequently, many studies focus on relatively low-resolution locomotor assays such as monitoring the average speed of a population, walking trajectories, or the ability to fulfill simple motor tasks such as climbing a vertical surface (*Ganetzky and Flanagan, 1978*; *Branson et al., 2009*; *Slawson et al., 2009*; *Robie et al., 2010*). To obtain kinematic data, researchers have relied on the manual frame-by-frame analysis of videos (e.g. *Strauss and Heisenberg, 1990*, *1993*; *Wosnitza et al., 2012*). The lack of a high-resolution and accessible assay to monitor fruit fly walking has greatly limited the use of this model system to study locomotion.

In order to quantitatively analyze locomotion in *Drosophila*, we developed an optical system to monitor walking that uses frustrated Total Internal Reflection (fTIR) coupled with high-speed video imaging. Our fTIR-based method is similar in principle to that used for analyzing the gaits of larger animals (e.g. CatWalk (*Vrinten and Hamers, 2003*); http://www.noldus.com/animal-behavior-research/products/catwalk/), but differs significantly due to the small size and rapid walking speed of fruit flies. Using this approach, we are able to track the position of each footprint relative to the body at high spatial and temporal resolution as an untethered fly walks freely on a flat surface. Custom analysis software allows the extraction of many parameters of fly walking behavior, including gait, coordination between legs, and footprint positions. Using this method, we comprehensively characterized the walking behavior of wild type animals. Unlike walking in vertebrates, our data demonstrate that flies do not abruptly switch from one gait to another, but instead rely on a continuum of gait patterns that correlate with walking speed. However, several readouts suggest that flies may use distinct neural programs at slow, medium, and fast walking speeds. Genetic manipulations to specifically disrupt sensory feedback from the legs show that blocking proprioception results in altered step parameters and reduced walking precision, especially at slower speeds, but does not interfere with the ability of flies to execute a tripod gait. Together, these data reveal the underlying parameters of wild type walking in *Drosophila* and show that proprioceptive sensory feedback is important, but not absolutely required, for coordinated locomotion.

## Results

### High resolution tracking of gait parameters

The analysis of locomotion in large animals, notably mammals, often relies on the placement of visual marks in strategic positions, usually joints that can be readily detected and tracked (*Akay et al., 2006*). However, in smaller insects such as *Drosophila*, such a strategy becomes not only technically challenging but is also likely to disturb walking behavior and generate artifacts. Such challenges have precluded a more detailed examination of the components that comprise the fly's walking behavior. To overcome these obstacles, and to measure the biomechanical features underlying walking in *Drosophila* we turned to an optical effect known as *frustrated Total Internal Reflection* (fTIR) (*Zhu et al., 1986*). Total Internal Reflection occurs when light traveling through a medium—in this case optical glass—hits an interface with another medium with a lower refractive index, such as air. If the angle of incidence is above the so-called critical angle (as compared to the normal of the surface), defined by Snell's Law (*Katz, 2002*), the light is no longer refracted but is internally reflected. For a glass-air interface this corresponds to ~43°. However, if a denser material, such as the tarsus of an insect leg, contacts the surface of the glass, then the locally 'frustrated' total internal reflection will scatter the light, which can be recorded by a high-speed video camera (*Figure 1* and *Figure 1—figure supplement 1*) (*Sumriddetchkajorn and Amarit, 2006*). A sample video of the unprocessed fTIR effect can be seen in *Video 1*.

To automate the tracking of the footprints and fly body revealed by the fTIR method, we created a program called FlyWalker that tracks and outputs several user-defined parameters (*Figure 1—figure supplement 2*). The program, which is freely available for download at http://biooptics.markalab.org/FlyWalker, evaluates the fTIR signals in each video and generates a set of graphs that describe the walking behavior (*Video 2* shows a FlyWalker-processed video; see 'Materials and methods' for a list of parameters, their definitions and a list of graphs generated by FlyWalker). In addition, the program also creates a spreadsheet that contains the full data set. From this, the user can analyze, compile and compare several samples.

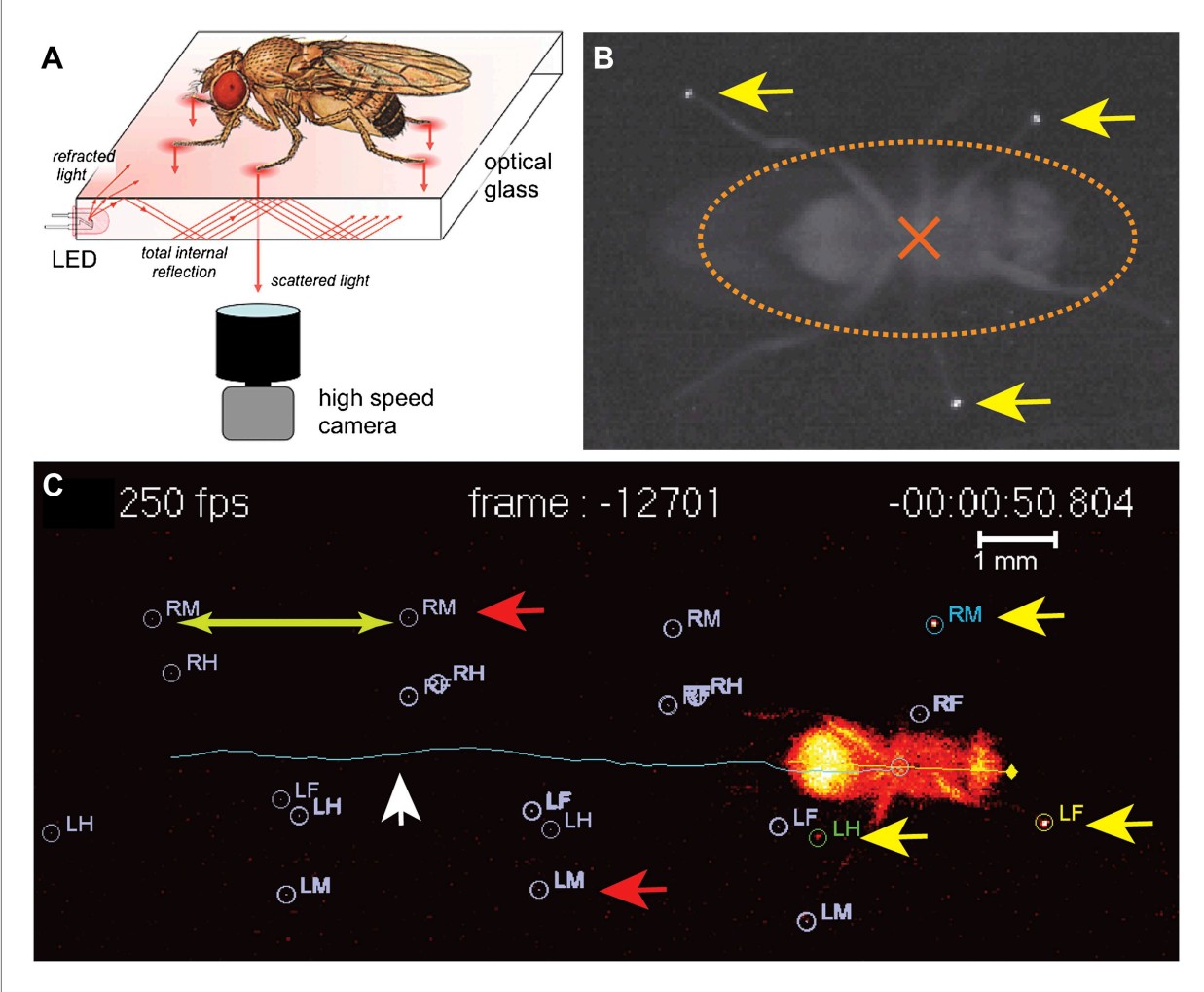

**Figure 1**. fTIR apparatus and FlyWalker software. (**A**). Schematic of the fTIR optical effect. LED light sources are located at the edges of an optical glass and light propagates within the glass via internal reflection. Tarsal contacts lead to light scattering detected by a high-speed camera. See *Figure 1—figure supplement 1* for more details. (**B**). Single frame of a fTIR video. The fTIR effect can be seen for three legs in stance phase (yellow arrows). Background light partially illuminates the fly's body (orange dashed ellipse; the center of the body is indicated by an orange cross). (**C**). Image generated by the FlyWalker software. The fly's footprints and body center are tracked throughout the video. Present footprints are identified and labeled (yellow arrows). The fly body and trajectory are visualized by a blue line (white arrow). Past footprints can also be recorded (red arrows). A scale bar can be introduced. *Step length* is defined as the distance between two consecutive footprints (green arrows).

The following figure supplements are available for figure 1:

**Figure supplement 1**. Additional information on fTIR apparatus.

**Figure supplement 2**. FlyWalker program.

## Step parameters

As a proof-of-principle, we examined the walking behavior of freely walking upright wild-type adult flies on a flat horizontal surface. The distance recorded was on average 1.27 cm, a distance covered by the camera without compromising the fTIR signal. We collected 71 videos of animals that walked in a straight manner without any stops. *Average speeds* of each fly varied between 7.2 and 44.7 mm/s with 28 mm/s as the most representative speed (*Figure 2A*), similar to previously reported values (*Robie et al., 2010*). Interestingly, speeds ~20 mm/s were underrepresented in this data set (see 'Materials and methods'), reminiscent of underrepresented speeds at gait transitions in humans or ponies (*Hoyt and Taylor, 1981*).

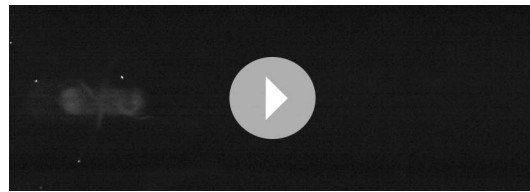

**Video 1**. Unprocessed fTIR video.

To initially describe the walking behavior of wild type animals, we plotted several step parameters (*Figure 2B—F* and *Figure 2—figure supplement 1*). As previously noted in several insect species (*Wilson, 1966*; *Graham, 1972*; *Strauss and Heisenberg, 1990*) as speed increases, *stance phase duration* becomes shorter while *swing phase duration* remains largely constant; at the fastest speeds the durations of both swing and stance phases equalize (*Figure 2B*). Consequently, *step period* varies inversely with *average speed* up to approximately 30 mm/s, when *step period* reaches a steady state of about ~60 ms, corresponding to a frequency of 16 cycles per s (*Figure 2C*).

Next we examined the relationship between step pattern and period for faster and slower walking animals. For this, we plotted the *metachronal lag*, (i.e., the time between sequential swing onsets from hind to forelegs on an ipsilateral side), as a function of hindleg *period* (*Graham, 1972*) (*Figure 2D*, *Figure 2—figure supplement 1*). We compared data extracted from slow animals (<20 mm/s) with those from fast animals (>34 mm/s). These speed groups were chosen and consistently used in this and all subsequent analyses because several of our readouts exhibit discontinuous or non-linear behavior at ~20 and ~34 mm/s (see *Figures 2A; 3C and 5B* for examples). Data extracted from the fast set reveal that hindleg *period* equals the *metachronal lag* ($_HLag_F ≈ P_H$), typical of the tripod gait (also called Gait I; (*Graham, 1972*)). In contrast, flies walking slower than 20 mm/s showed a *period*-dependent *metachronal lag* with a regression curve of ($_HLag_F = 0.505* P_H + 27.6$), typical of the tetrapod gait (also called Gait II; (*Graham, 1972*)), see also 'Gait parameters', below) (*Figure 2D*). We also find that *step length* increased almost linearly with speed (*Figure 2E*). As a consequence, *swing speed* also follows this trend (*Figure 2F*).

The FlyWalker program also extracts data for each individual leg for each of the three thoracic segments (*Figure 2—figure supplement 2*). Although legs in different segments show very similar trends for most parameters, some distinctions can be made. For example, as *average speed* increases there is a small but noticeable decrease in *swing phase duration* in the forelegs when compared to other segments. In addition, compared to other leg segments, foreleg *swing speed* varies more linearly with *average speed*, as seen by a larger R value (*Figure 2—figure supplement 2*). These observations support the hypothesis of a leading role for forelegs in forward locomotion suggested by experiments in the stick insect (*Akay et al., 2007*; *Bässler and Büschges, 1998*).

## Spatial parameters

In addition to following the six footprints, FlyWalker also monitors the position of the fly body. From these data we can reconstruct the stance positions relative to the center of the body (*Figure 3A*). In order to account for variations in body size, these plots are normalized to body length. Stance traces are generated as each leg contacts the glass. The onset of these traces is termed the *Anterior Extreme Position* (*AEP*), which corresponds to the position where the leg first contacts the glass after touchdown at the end protraction (*Cruse, 1976*). The position at the end of the stance phase, just before the tarsi enter swing phase, corresponds to the *Posterior Extreme Position* (*PEP*) (*Cruse, 1976*). *Stance traces* can be compared by their straightness, which is a measure of how much wobble there is in body position relative to each footprint, which are stationary. This parameter, the *stance linearity index*, is calculated by computing the average difference between an actual stance trace and a smoothed version of the trace (*Figure 3B*, *Figure 3—figure supplement 1*). Measuring this parameter, faster animals have straighter stance traces (lower *stance linearity indexes*) compared to slower animals. This trend is seen up to approximately 34 mm/s, after which this value stabilizes (*Figure 3C*).

Another informative parameter corresponds to the clustering of the *AEP*s and *PEP*s for each

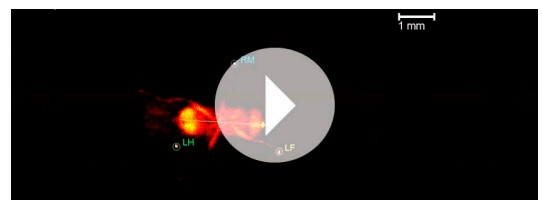

**Video 2**. Processed video by FlyWalker.

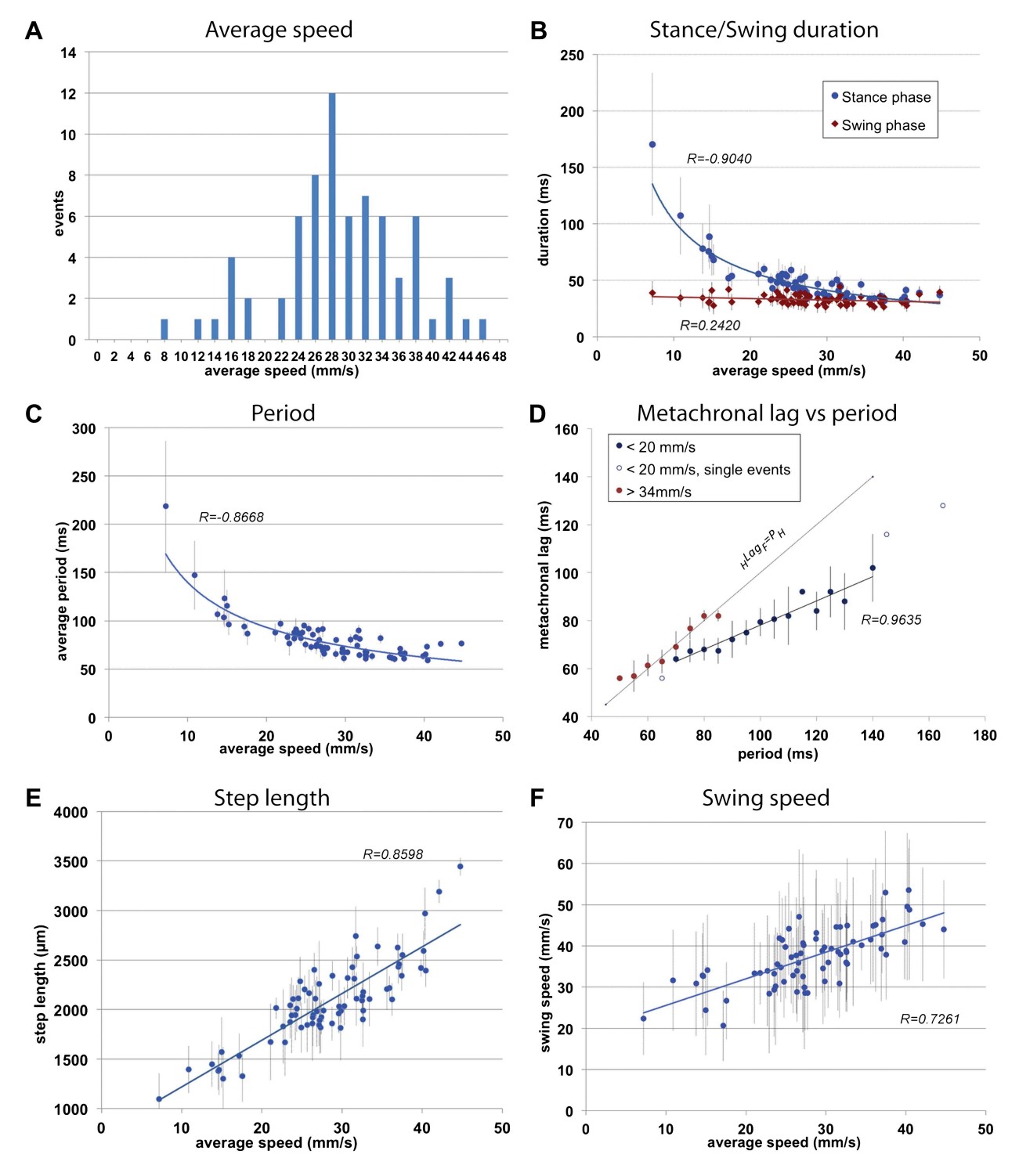

**Figure 2**. General walking parameters. (**A**). Speed histogram of 71 videos recorded for wild type flies, with 2 mm/s bins. Average speeds vary between 7.2 and 44.7 mm/s, with 28 mm/s the most represented speed. Speeds at ~20 mm/s are underrepresented in this dataset. (**B**). Durations of *stance* (blue) and *swing* (red) *phases* as a function of speed. *Swing phase* duration remains mostly constant while stance phase duration is inversely proportional to speed. Graphical fits for *swing* and *stance phases* versus speed are represented in red and blue lines, respectively. (**C**). *Step period* is inversely proportional to average speed. The blue line is a graphical fit. (**D**). Duration of the *metachronal lag* as a function of the hindleg *period* for slow (blue) and fast (red) flies. *metachronal lag* closely matches hindleg *period* for fast flies ($_H$Lag$_F$ ≈ Period). Regression line for the slow walking flies: $_H$Lag$_F$ = Period × 0.505 + 27.627. A total of 220 metachronal waves were used, error bars correspond the standard error of the mean. (**E**),(**F**). Step

*Figure 2. Continued on next page*

*Figure 2. Continued*

*length* (**E**) and *swing speed* (**F**) increase linearly with speed. A graphical fit is shown by the blue lines. Error bars in (**B**–**F**) correspond to standard error of the mean.

The following figure supplements are available for figure 2:

**Figure supplement 1**. Gait definitions.

**Figure supplement 2**. Gait parameters by segment.

leg in a single video. This parameter—termed *footprint clustering*—corresponds to the vector sum of standard deviations (STDs) from the mean for all *AEP*s or *PEP*s calculated for each leg (*Figure 3D*). For example, a small *footprint clustering* value for the foreleg *AEP* means that the *AEP* coordinates (relative to the fly's body) were similar for all of the foreleg steps in a video. Using this parameter, we asked if there was a relationship between *footprint clustering* and *average speed*. For the reasons given above, we binned the speed values into three groups: <20 mm/s; between 20 and 34 mm/s; and >34 mm/s. For both *AEP* and *PEP*, the footprint clustering values were smaller for faster flies (*Figure 3E* and *Figure 3—figure supplements 2 and 3*), suggesting that animals have more spatially restricted steps as they increase their speed. We also found that *AEP* clustering values are generally smaller than *PEP* clustering values (*Figure 3E*), suggesting a tighter motor control at touchdown (*AEP*) compared to stance offset (*PEP*). Pooling the data for all legs, *AEP* clustering was smaller than *PEP* clustering in 61 out of 71 (86%) videos. This difference is largest for the forelegs and is very small for the hindlegs (*Figure 3—figure supplement 3*).

Knowing that flies increase their speed by making longer strides (*Figure 2E*), we next asked if there is a relationship between *average speed* and *AEP* or *PEP*. To address this question we plotted *AEP* and *PEP* for each leg for the three speed groups (*Figure 3F*). At faster speeds, for all legs, *AEP* coordinates are shifted anteriorly, while *PEP* coordinates are shifted posteriorly. Interestingly, at faster speeds midleg *AEP* and *PEP* are also shifted laterally, perhaps to increase stability at higher speeds. In contrast, the hindlegs are positioned closer to the body at higher speeds, perhaps to allow for a stronger power stroke.

## Gait parameters

Previous work described insect gaits as either tripod or tetrapod, depending on the speed and body load (*Graham, 1972*). The *tripod gait* is characterized by three legs in stance phase and three legs in swing phase at any one time (*Figure 4—figure supplement 1*). Each group of three legs is composed of the fore and hind legs on one side and the midleg on the contralateral side. In contrast, in an idealized *tetrapod gait* only two legs are in swing phase while the remaining four legs are in stance phase (*Figure 4—figure supplement 1*). The two legs in swing phase are on contralateral sides and are offset by one segment. We also observed many noncanonical stance combinations that do not fit either of the idealized gaits, which we analyze below (*Figure 4—figure supplement 1* and *Table 1*).

For each video, a step pattern can be generated while simultaneously plotting the *instantaneous speed* and gait characteristics with high temporal resolution (*Figure 4A,B*). Notably, the *instantaneous speed* plot has a wave-like appearance (*Figure 4A',B'*), particularly in high speed animals, with maxima and minima that can differ up to 30 mm/s. Consistent with the observations of Graham in the first instar stick insect (*Graham, 1972*), peak speeds are observed midway through each stance phase when the retracting tripod reaches its maximum motor output. Conversely, minimum speeds are observed at the transition between phases, when the stance switches to a different set of legs. The difference between the maxima and minima in these *instantaneous speed* plots decreases at slower speeds (*Figure 4B*), suggesting that power is more evenly distributed throughout each period at slower speeds.

For each frame in a video we also classify whether the fly is in a tripod, tetrapod, or noncanonical stance. The resulting *gait map* graphically illustrates the gaits used over time (yellow for tripod, blue for tetrapod, and grey for noncanonical) (*Figure 4A",B"*). Visual inspection of the full data set shows that flies walk preferentially using the tripod gait (data not shown) (*Strauss and Heisenberg,*

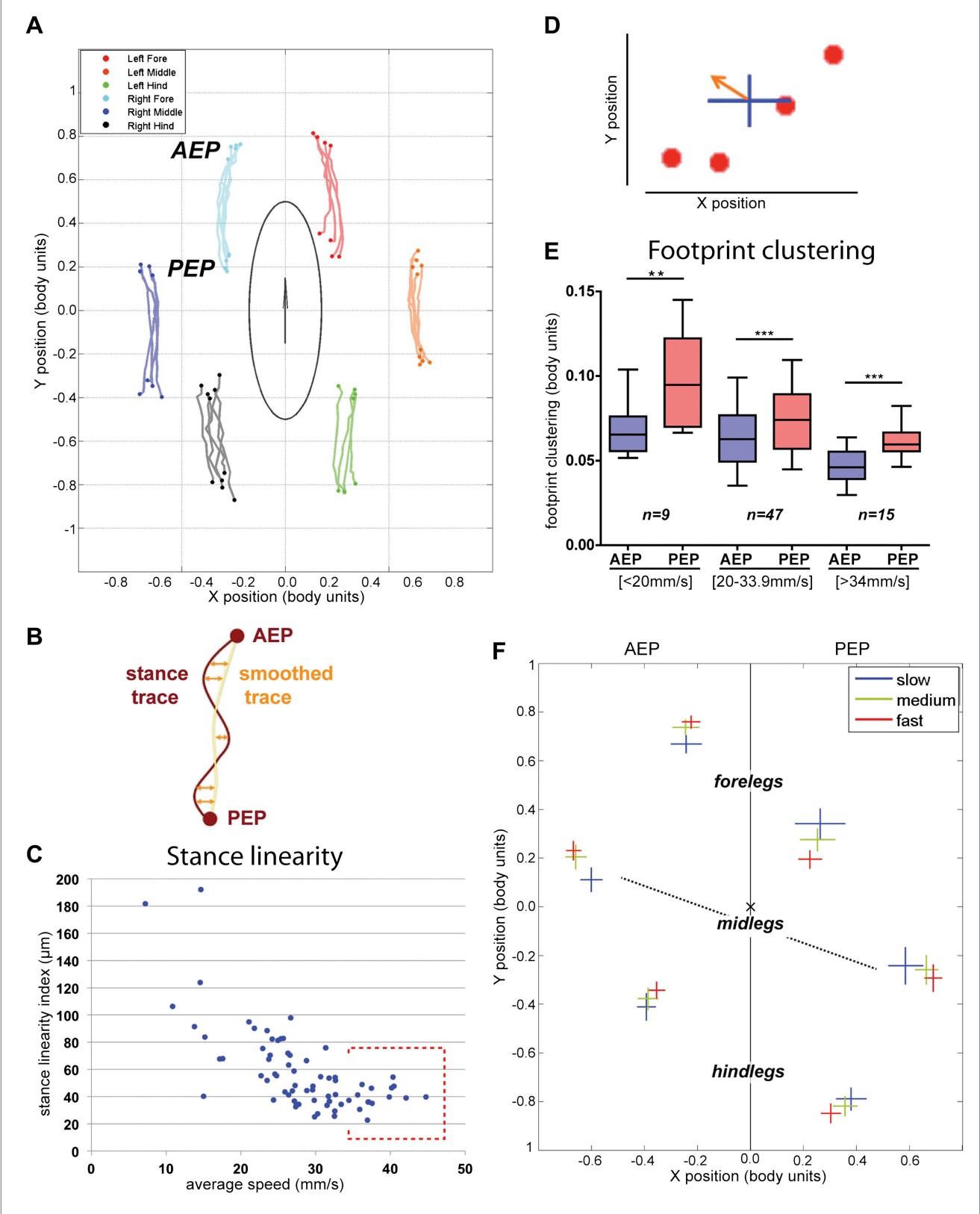

Figure 3. Spatial parameters. (**A**). *Stance traces.* Representative plot of an animal walking at 28.82 mm/s. Traces are generated by the position of the stance phase footprints relative to the body center (set at 0.0,0.0). For each leg, stance onset corresponds to the *Anterior Extreme Position* (*AEP*) while
*Figure 3. Continued on next page*

*Figure 3. Continued*

stance offset is termed *Posterior Extreme Position* (*PEP*). (**B**). Method to quantify the *stance linearity index*. For each stance trace (brown), a smoothed trace is generated (using data from every five frames; yellow trace), and the average of the difference between these two lines (orange arrows) corresponds to the *stance linearity index*. (**C**). *Stance linearity* as a function of speed. Each data point corresponds to the average of all traces for all six legs for a single video greater than ~34 mm/s, *stance linearity* becomes constant (red box). (**D**). Method to quantify *footprint clustering*. For each set of *AEP/PEP* footprints (red circles), an average ±STD xy point is created (blue cross). The *footprint clustering* value is calculated as the vector sum of the two STD values (orange arrow). (**E**). Quantification of *footprint clustering*. Data were grouped into slow (<20 mm/s), medium (between 20 and 34 mm/s) and fast (>34 mm/s) speeds. Boxplots represent the median as the middle line, with the lower and upper edges of the boxes representing the 25% and 75% quartiles, respectively; the whiskers represent the range of the full data set. Values are normalized for body size. Asterisks indicate the significance of the decrease in *footprint clustering* between the *AEP* and the *PEP* (using the paired parametric t test in the case of the slow speed group and the Wilcoxon non-parametric test for the remaining groups, **p<0.005, ***p<0.001). A comparison between the three speed groups also displays statistical significance (Kruskal–Wallis-ANOVA, p values of 0.0015 and 0.0014 for *AEP* and *PEP*, respectively). Dunn's *post hoc* significance tests show: slow *AEP* vs medium *AEP*, not significant (NS); medium *AEP* vs fast *AEP*, **; slow *AEP* vs fast *AEP*, **; slow *PEP* vs medium *PEP*, NS; medium *PEP* vs fast *PEP*, NS; slow *PEP* vs fast *PEP*, ***. (**F**). Footprint position relative to the body center. Data were pooled as in the previous panel. *AEP* and *PEP* values for each leg are represented on the left and right sections of the plot, respectively. Values are normalized for body size.

The following figure supplements are available for figure 3:

**Figure supplement 1**. Representative examples of stance traces and corresponding stance linearity values.

**Figure supplement 2**. AEP and PEP clustering values for all segments as a function of speed.

**Figure supplement 3**. AEP and PEP clustering values for each segment as a function of speed.

*1990*; *Wosnitza et al., 2012*), but also underscores that as flies decrease their speed they increasingly use tetrapod and noncanonical combinations. To quantify these *gait maps* we plot the *gait index*. To calculate the *gait index* each frame is assigned a value (+1 for tripod, −1 for tetrapod and 0 for noncanonical) and these values are averaged for a sliding window of *n* frames; empirically, we find that *n* = 8 is most effective at distinguishing flies primarily using the tetrapod and tripod gaits (*Figure 4A‴,B‴* and *Figure 4—figure supplement 2*). The average value for an entire video corresponds to the *average gait index*. We calculated the *average gait index* for all 71 videos in our data set and binned the results using the same three speed groups used previously (*Figure 4C*). Each group (slow, medium, and fast) had statistically distinct *average gait indexes*: slower animals had more negative values while faster animals had increasingly positive values, reflecting their increasing use of the tripod gait.

As a complementary method to quantify differences in gait, we analyzed our fTIR data to determine the fraction of time flies spend using either idealized tripod or tetrapod configurations, and if these fractions depend on *average speed* (*Figure 4D,E*). We define the *tripod* or *tetrapod indexes* as the fraction of frames in a video in which a leg combination matches one of these two gaits. As expected, faster flies spend a larger fraction of time in a tripod configuration, which presumably allows flies to maximize leg thrust (*Figure 4D*). The inverse relationship is seen for the tetrapod configuration, where slower animals display the highest proportion of this gait (*Figure 4E*).

Two additional conclusions can be derived from these data. First, there is not an abrupt transition between the tripod and tetrapod gaits at a particular speed. Instead, the proportion of time a fly spends using each gait varies gradually according to *average speed*. Second, the idealized tripod and tetrapod combinations only account for a subset of all stance configurations present in our data set. For example, the fastest flies in our data set had *tripod* and *tetrapod indexes* of ~70% and <10%, respectively. Noncanonical leg configurations in which none or only one leg is in swing phase account for 48% to 21% of all combinations, depending on the *average speed* (*Table 1*). Inspection of these videos reveals that these noncanonical combinations generally occur at the transitions between stances (data not shown and *Figure 4A″*). Thus, slower animals spend less time using the tripod gait in part because they use the tetrapod gait more, but also because they spend a larger fraction of time in gait transitions, where noncanonical leg configurations play a dominant role. In addition, at slow speeds occasional short sequences of pentapod stances (≤3 steps) fit the so-called '*wave gait*' in which individual legs swing in a wave-like pattern from front to back (*Wosnitza et al., 2012*).

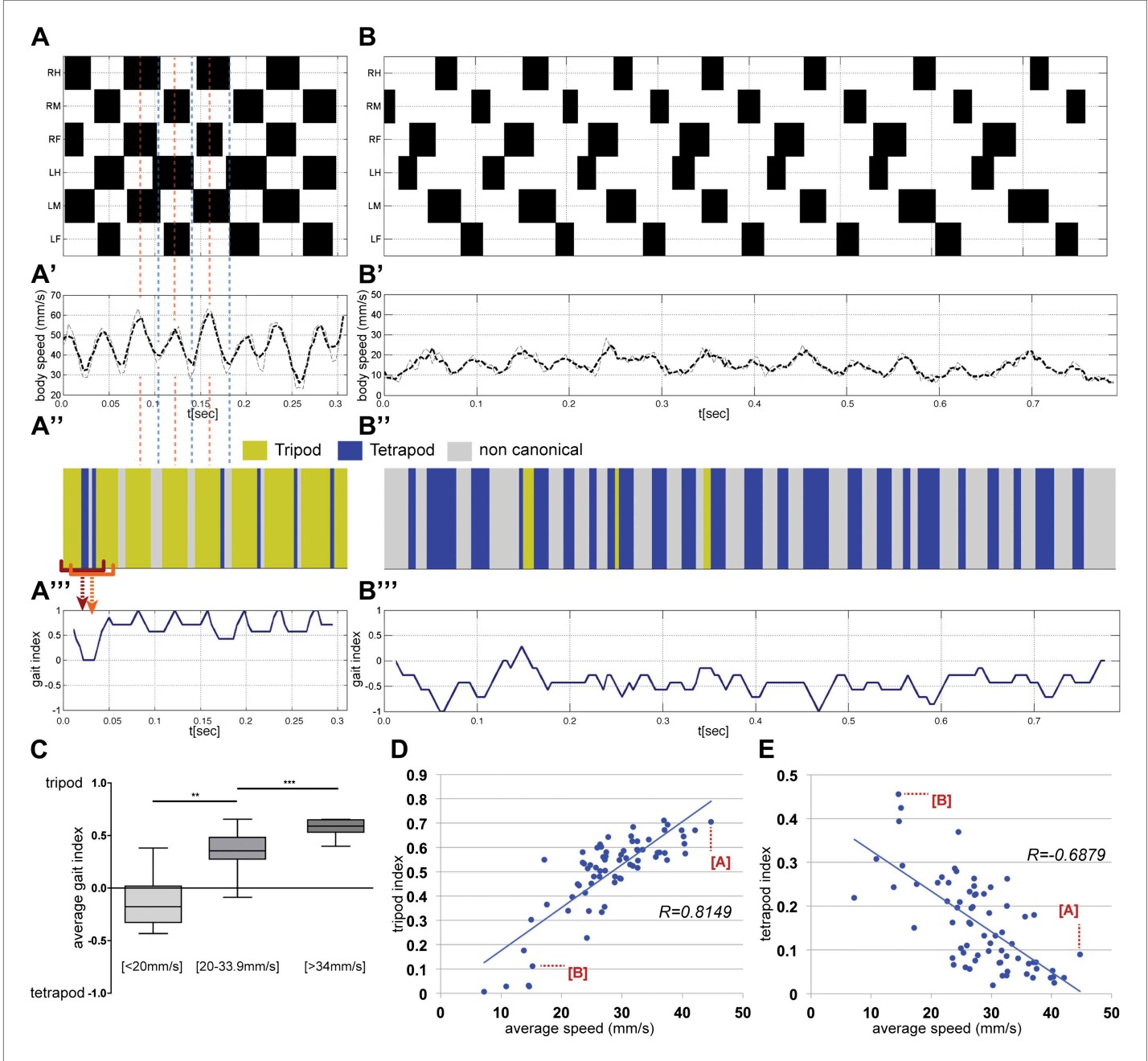

**Figure 4**. Gait parameters. (**A**),(**B**). Upper panels show the step pattern for representative videos of animals walking at 44.7 (**A**) and at 14.6 mm/s (**B**). For each leg swing phases are represented in black (from top to bottom: right hind (RH); right mid (RM); right front (RF); left hind (LH); left middle (LM); left front (LF)). (**A'**),(**B'**) show the *instantaneous speed* for the same video. Thick and thin lines correspond to integration times of 25 or 12.5 ms, respectively. In faster animals (**A**), peak speeds are observed halfway through the stance phase (red dashed line), while minimum speeds are observed during gait transitions (blue dashed line). (**A''**),(**B''**), for each frame the corresponding gait was color coded as follows: green (tripod), blue (tetrapod), and gray (non-canonical). Red brackets indicate the eight frame windows used to generate the gait index plots (**A'''**),(**B'''**). Tripod = +1; tetrapod = −1; and noncanonical = 0. (**C**). Quantification of the *average gait index* for three speed groups. For each video the average gait index was calculated for all frames (p<0.0001 for Kruskal–Wallis-ANOVA test. Dunn's *post hoc* significance test: **p<0.005, ***p<0.001). (**D**),(**E**). Tripod (**D**) and tetrapod (**E**) indexes as a function of speed. Graphical fits are represented in blue. Data for points labeled (**A**) and (**B**) are shown in panels (**A**) and (**B**), respectively.

The following figure supplements are available for figure 4:

**Figure supplement 1**. Gait features.

**Figure supplement 2**. Gait index plot.

**Table 1.** Gait combinations in three speed classes[a]

| Slow [≤19.9 mm/s] | | Medium [20–33.9 mm/s] | | Fast [≥34 mm/s] | |
|---|---|---|---|---|---|
| Tripod | 31.37 | Tripod | 51.63 | Tripod | 64.98 |
| Tetrapod | 25.45 | Tetrapod | 15.90 | Tetrapod | 7.26 |
| *Total* | *56.82* | *Total* | *67.54* | *Total* | *72.24* |
| *Additional combinations* | | *Additional combinations* | | *Additional combinations* | |
| 111110 | 8.27 | 111111 | 4.76 | 111011 | 4.89 |
| 110111 | 7.60 | 111110 | 4.61 | 111111 | 4.25 |
| 111111 | 6.38 | 011111 | 4.30 | 011111 | 4.07 |
| 111011 | 5.62 | 111011 | 4.26 | 010111 | 3.36 |
| 011111 | 5.03 | 010111 | 3.84 | 111010 | 2.80 |
| 010111 | 2.47 | 110111 | 3.79 | 110111 | 2.32 |
| 111010 | 2.45 | 111010 | 2.91 | 111110 | 1.29 |
| 101111 | 1.94 | 101111 | 0.71 | 011011 | 1.25 |
| 111101 | 1.69 | 011011 | 0.52 | 011010 | 0.69 |
| Pentapod | 30.15 | Pentapod | 14.37 | Pentapod | 13.33 |
| 3 gaits combined[b] | 86.97 | 3 gaits combined[b] | 81.91 | 3 gaits combined[b] | 85.57 |
| *Total*[c] | *98.3* | *Total*[c] | *97.2* | *Total*[c] | *97.15* |

[a]Values are expressed as percentage (%). Leg order in combination: LF LM LH RF RM RH. 1, footprint present; 0, footprint is absent.
[b]The sum of tripod, tetrapod, and pentapod gaits.
[c]The sum of all gait patterns listed in the table.

## Coordination parameters

Typically for hexapods, leg touchdown occurs close to where the immediately anterior ipsilateral leg made contact, a behavior termed *follow-the-leader* (***Dean and Wendler, 1983***; ***Cruse et al., 1984***; ***Song and Choi, 1989***; ***Brunn and Dean, 1994***; ***Sponberg and Full, 2008***). This behavior depends on sensory feedback and specialized intersegmental interneurons (***Brunn and Dean, 1994***). One possible advantage of this behavior is to ensure that the animal places its legs on safe ground, particularly on rough terrain. As a consequence, mid and hindleg footprints fall close to where the foreleg was placed (***Figure 1C***). We quantified this tendency by measuring the *footprint alignment* parameter, which corresponds to the mean standard deviation of the projection of the fore, mid, and hind footprints along the body's displacement axis (***Figure 5A***). Accordingly, footprints that are more aligned have a smaller *footprint alignment* value (***Figure 5—figure supplement 1***). Interestingly, plotting *footprint alignment* as a function of speed revealed a non-linear relationship in which the data points cluster into three groups (***Figure 5B***). At slow speeds (<20 mm/s), *footprint alignment* values were relatively large while at fast speeds (>34 mm/s) these values were much smaller, suggesting that *footprint alignment* is highly constrained. Notably, at intermediate speeds (between 20 and 34 mm/s), there was no correlation between speed and *footprint alignment*. The more aligned footprints observed in faster flies could assist the maintenance of static stability during tripod transitions (***Song and Choi, 1989***). In support of this possibility, the *tripod index* increases with lower *footprint alignment* values (data not shown). The abrupt difference seen in alignment values between slow, intermediate, and fast flies suggests that the motor circuits controlling walking in these three speed groups may differ. Interestingly, one of the transitions in *footprint alignment* values occurs at speeds (~20 mm/s) that are underrepresented in our data set, consistent with the existence of a possible gait transition at this speed.

Coordinated walking requires that CPGs interact with other CPGs. Such coordination is seen when examining the phase differences between the contralateral and ipsilateral legs of the same segment. For example, during a tripod gait, contralateral legs within the same segment maintain a consistent phase value of 0.5 ± 0.05 (***Graham, 1972***). The FlyWalker software computes the phase values for contralateral legs within the same segment and between adjacent ipsilateral legs. We compared

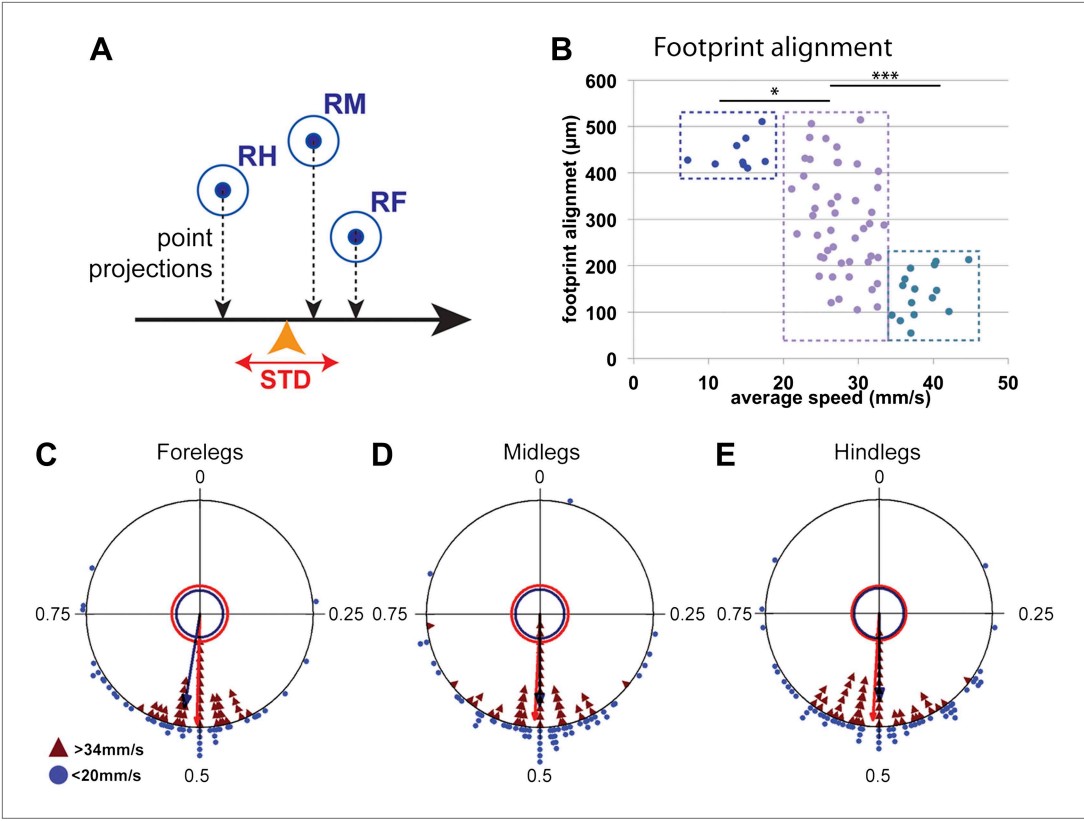

**Figure 5**. Coordination parameters. (**A**). Method to calculate *footprint alignment*. For each set of footprints, the projection points along the displacement axis (horizontal black arrow) are calculated. *Footprint alignment* corresponds to the standard deviation (STD) of the average point (orange arrowhead). (**B**). Quantification of *footprint alignment* versus speed. Values are color coded according to speed group: <20 mm/s in dark blue; between 20 and 34 mm/s in purple and >34 mm/s in turquoise. Values cluster into three main groups indicated by the dotted boxes (p<0.0001 for Kruskal–Wallis-ANOVA test. Dunn's post hoc significance test: *p<0.005, ***p<0.001). (**C**)–(**E**). Radial plots of contralateral leg phases comparing slow (blue triangles) and fast (red triangles) walking animals. Mean r vectors for slow and fast animals are represented by blue and red arrows, respectively. Inner circles indicate a Rayleigh p value of 0.05.

The following figure supplements are available for figure 5:

**Figure supplement 1**. Representative images of footprints with different footprint alignment values.

**Figure supplement 2**. Radial plots of adjacent ipsilateral leg phases comparing slow and fast walking animals.

these values for the slow (<20 mm/s) and fast (>34 mm/s) groups of flies in our data set. In both groups, contralateral legs within the same segment display an average phase of approximately 0.5 (*Figure 5C–E*). However, slower animals display a higher degree of variability as seen by a shorter r vector in these radial plots. Examining hindlegs vs midlegs and midlegs vs forelegs revealed a decrease in the phase values for slower animals (*Figure 5—figure supplement 2*), consistent with an increased step period without a proportional increase in the step lag.

## Consequences of impairing leg sensory feedback

The mechanosensory system constantly reports the surface properties and the relative position of each of an animal's appendages (*Bässler, 1977*). Moreover, sensory feedback is thought to trigger inter-leg coupling of local circuits to allow stable and well-coordinated gaits (*Brunn and Dean, 1994*). Different types of proprioceptor organs, positioned in different regions of the legs, report different aspects of the posture and terrain. For example, the tibial campaniform sensilla (or sensilla campaniformia) are

mostly responsible for measuring body load while the femoral chordotonal organ (ChO) is a stretch sensor, reporting joint angles as the animal walks (*Shanbhag et al., 1992*; *Zill et al., 2004*).

To quantify the effect of inactivating sensory neurons in the *Drosophila* leg, we tested both inactivation of a small subset of neurons and broader sensory inactivation within the legs. First, we tested the inactivation of the leg ChO. The *nanchung* (*nan*) gene encodes for a cation channel subunit and is expressed exclusively in the sensory cilia of chordotonal organs (*Kim et al., 2003*) (*Figure 6A*). Loss of function alleles for *nan* display loss of sound perception, hygrosensation and negative gravitaxis defects, in addition to an 'uncoordinated' phenotype (*Liu et al., 2007*; *Kamikouchi et al., 2009*; *Sun et al., 2009*). Second, we tested neuronal inactivation induced by the expression of tetanus toxin (TNT) in a large subset of sensory neurons in the legs (*Figure 6B*). To restrict TNT expression to leg sensory neurons we relied on a intersectional approach using a *UAS-FRT-stop-FRT-TNT* line (*Stockinger et al., 2005*), the pan-sensory driver *5-40-Gal4* (*Hughes and Thomas, 2007*), and a 567 base pair *cis*-regulatory enhancer fragment of *dachshund* (*dac*) to drive the FLP recombinase in the leg imaginal disc (*Giorgianni and Mann, 2011*). For simplicity, we refer to this genotype as *5-40$^{Leg}$>TNT*. *Video 3* shows an example of walking by these flies.

Flies of both genotypes (*nan$^{36a}$* and *5-40$^{Leg}$>TNT*) walked slower than wild type flies but, remarkably, maintained a typical tripod gait (*Figure 6C,D*). These flies also exhibited normal left-right and intersegmental coordination (*Figure 6E*). Strikingly, however, stance traces for *nan$^{36a}$* and *5-40$^{Leg}$>TNT* flies highlight several locomotion defects (*Figure 7A*), which were quantified (*Figure 7* and *Table 2*). We observed an increase in the *step length* (*Figure 7B*); a more wobbly body placement (reflected by an increase in *stance linearity*, *Figure 7C*); and a larger variability in *footprint clustering* (*Figure7D*). In addition, both *AEP* and *PEP* were altered, in part because of a longer stride, and midleg placement was farther from the body (*Figure 7A* and data not shown). Both *swing* and *stance duration* increased (*Figure 7E,F*) resulting in a longer *period*. The increase in *step length* reflects both overreaching (more anterior AEPs) and delayed swing onset (more posterior PEPs). However, *swing speeds* were minimally affected (*Figure 7G*), suggesting that the motor neurons, themselves, were not compromised in proprioception deficient flies. *Footprint alignment* values also decreased in flies deprived of sensory feedback (*Figure 7H*). This phenotype was partially a consequence of a longer stride but also because protraction was often completed when direct contact was made with the next anterior leg. Importantly, although both *nan$^{36a}$* and *5-40$^{Leg}$>TNT* affect sensory structures in the antennae, only an increase in *swing duration* was observed upon surgical removal of the antenna, arguing that all other phenotypes are a result of knocking out sensory feedback specifically from the legs (data not shown).

Interestingly, several of the differences we observed in flies deprived of sensory feedback were more pronounced in animals that walked more slowly compared to faster flies (e.g. *stance duration* and *footprint alignment*; *Figure 7F,H*). The *tripod index* also decreased in slower flies (*Figure 6D*). It is also noteworthy that several of the trends observed in sensory-deprived flies (such as *AEP*, *PEP*, and *footprint alignment*) are the same as those observed in faster wild type flies. In addition, sensory deprived flies position their midlegs further from their bodies, similar to the trend seen in fast flies (*Figures 3F and 7A*). These observations are consistent with the idea that flies walking at slow, medium, and fast speeds use distinct neural programs, and that flies walking at fast speeds are less dependent on sensory feedback.

## Discussion

Research using insects has contributed greatly to the field of locomotion and sensory feedback (*Cruse, 1990*; *Cruse et al., 2007*; *Ritzmann and Büschges, 2007*; *Sponberg and Full, 2008*). Emerging and established tools have set the fruit fly as a valuable genetic system for studying many behaviors such as olfaction and courtship. Improved expression tools (*Pfeiffer et al., 2008*; *Jenett et al., 2012*; *Jory et al., 2012*; *Manning et al., 2012*), transgenic neuromodulators (*Pulver et al., 2009*) and imaging of neuronal activity (*Seelig et al., 2010*), continue to improve the *Drosophila* toolkit. However, despite the growing collection of tools to disturb the circuit that regulates waking, *Drosophila* neurogenetics lacks a reliable method to measure the outcome of these manipulations. Here we fill this gap in the toolkit by describing an approach that unambiguously detects with high temporal and spatial resolution the kinematic behavior of freely walking fruit flies on a flat surface. A powerful and downloadable FlyWalker software tool allows the tracking of both the fly body and legs when they contact the ground as the animal moves forward.

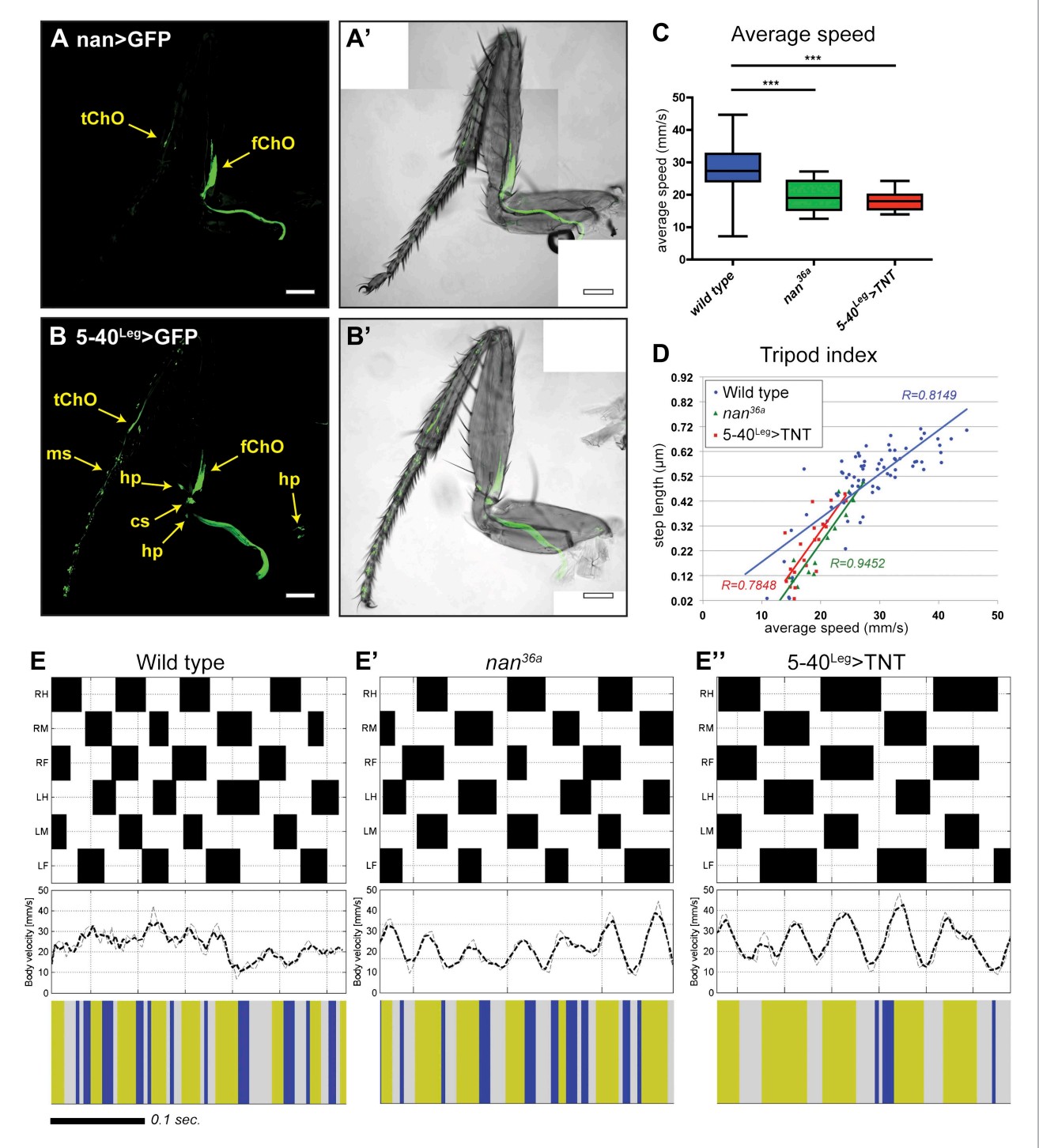

**Figure 6**. Effects of sensory deprivation on walking. (**A**). Expression pattern driven by *nanchung*-Gal4. Genotype: *F-Gal4, UAS-GFP*. (**B**). GFP expression under combinatorial control of *5-40-Gal4* and *dac^{RE}-flp*. Genotype: *5-40-Gal4, dac^{RE}-flp, UAS-FRT-stop-FRP-GFP*. All classes of sensory neurons in the leg express GFP. tChO, tibia chordotonal organ; fChO, femur chordotonal organ; hp, hair plates; cs, campaniform sensilla; ms, mechanosensory brisles. Bar, 100 μm. (**C**). *Average speed*. Boxplots with the median as the middle line and the lower and upper edges of the boxes representing the 25% and 75% quartiles, respectively; the whiskers represent the range of the full data set. Statistical analysis with one-way-ANOVA (p<0.0001) followed by Tukey's *post hoc* test, ***p<0.001. (**D**). *Tripod index*. Lines represent graphical fits. See **Table 2** for statistical analysis. (**E**). Gait patterns, instantaneous speeds, and gait maps for representative wild type and sensory deprived animals walking at similar speeds. See **Figure 2—figure supplement 1** for details. Tripod gait properties remain unchanged.

**Video 3**. Processed video of *5-40^Leg^>TNT* fly.

Using these tools, we analyzed the behavior of wild-type animals during straight walking on a horizontal plane, extending previous analyses (*Strauss and Heisenberg, 1990*, *1993*; *Wosnitza et al., 2012*). In addition to corroborating earlier findings, the high degree of temporal and spatial resolution, coupled to the simultaneous tracking of the body allowed us to define several additional parameters, particularly those that address the spatial aspects of locomotion. Several observations also suggest the existence of previously unknown transitions that occur at ~20 mm/s and ~34 mm/s. For one, we find that flies walking at 20 mm/s are underrepresented in our data set. In addition, some parameters, such as *stance linearity* and *footprint alignment*, show striking differences in flies walking slower and faster than these speeds. These transitions do not, however, represent abrupt transitions in gait because the tripod gait was observed at all speeds, at gradually diminishing frequencies in slower flies. This scenario contrasts with abrupt gait transitions observed in many vertebrates, for example horses, where each gait is only observed in a defined range of speeds (*Hoyt and Taylor, 1981*). Nevertheless, our data suggest that in flies modifications of a common neural circuit may regulate walking at these different speeds (*Watson et al., 2002*; *Sponberg and Full, 2008*). Because each leg muscle is targeted by multiple motor neurons (*Baek and Mann, 2009*), it is feasible that different subsets of MNs play a more or less dominant role at different speeds.

## Walking by proprioceptive-deficient flies

In larger insects such as the stick insect and locust, the role of sensory feedback in coordinated walking has been addressed by local surgical ablations coupled to electrophysiological measurements (*Usherwood et al., 1968*; *Cruse et al., 1984*; *Akay et al., 2001*; *Borgmann et al., 2009*). For example, using single leg preparations of the stick insect, it was found that for mid and foreleg steps to be out of phase (as they are in wild type walking), stimulation of sensory afferents in the midleg was required; without this stimulation, the CPGs for these legs fired in phase (*Borgmann et al., 2009*). These, as well as additional observations (*Büschges et al., 2008*), argue that sensory feedback from legs in stance phase is required for interleg coordination.

The use of *Drosophila* brings an additional set of tools to address these questions. In one set of experiments, we took advantage of a mutation that specifically disrupts the function of the ChOs while a second set of experiments used a combinatorial misexpression approach to inactivate the majority of sensory neurons in the leg, including the ChO, hair plates, campaniform sensilla and mechanosensory bristles. In both sets of experiments, gait parameters and interleg swing phases were largely normal when sensory feedback was impaired. These results suggest that in the fly interleg coordination is not dominated by sensory feedback. Instead, they suggest that communication between CPGs, perhaps by local interneurons, may be sufficient for coordination, a suggestion that is supported by recent studies in vertebrates ((*Kiehn, 2011*) for review). In insects, interneurons have been implicated in interjoint coordination within individual legs and in footfall placement, but their role in interleg coordination has not been established (*Brunn and Dean, 1994*; *Brunn, 1998*; *Matheson, 2002*). The different requirements for sensory feedback observed here compared to electrophysiology studies in the stick insect may in part be because in our experiments leg sensory neurons were inactivated in all six legs (in the case of *5-40^leg^>TNT*) or all ChOs in the animal (in the case of the *nan* mutant). We speculate that when all six legs are equally impaired, the flies resort to a CPG and interneuron-dominated system, which is sufficient to execute the tripod gait. Consistent with this notion, although interleg coordination was impaired upon amputation of a single hindleg in *Drosophila*, the flies were able to recover a partial tripod gait (*Wosnitza et al., 2012*).

The data derived from sensory-deprived flies also supports the idea that flies may be less dependent on sensory feedback when they walk fast because, for several parameters, the defects were significantly more pronounced in slow flies. This observation is consistent with the idea that when flies walk fast, they use a largely sensory-independent CPG-based system where individual CPGs communicate

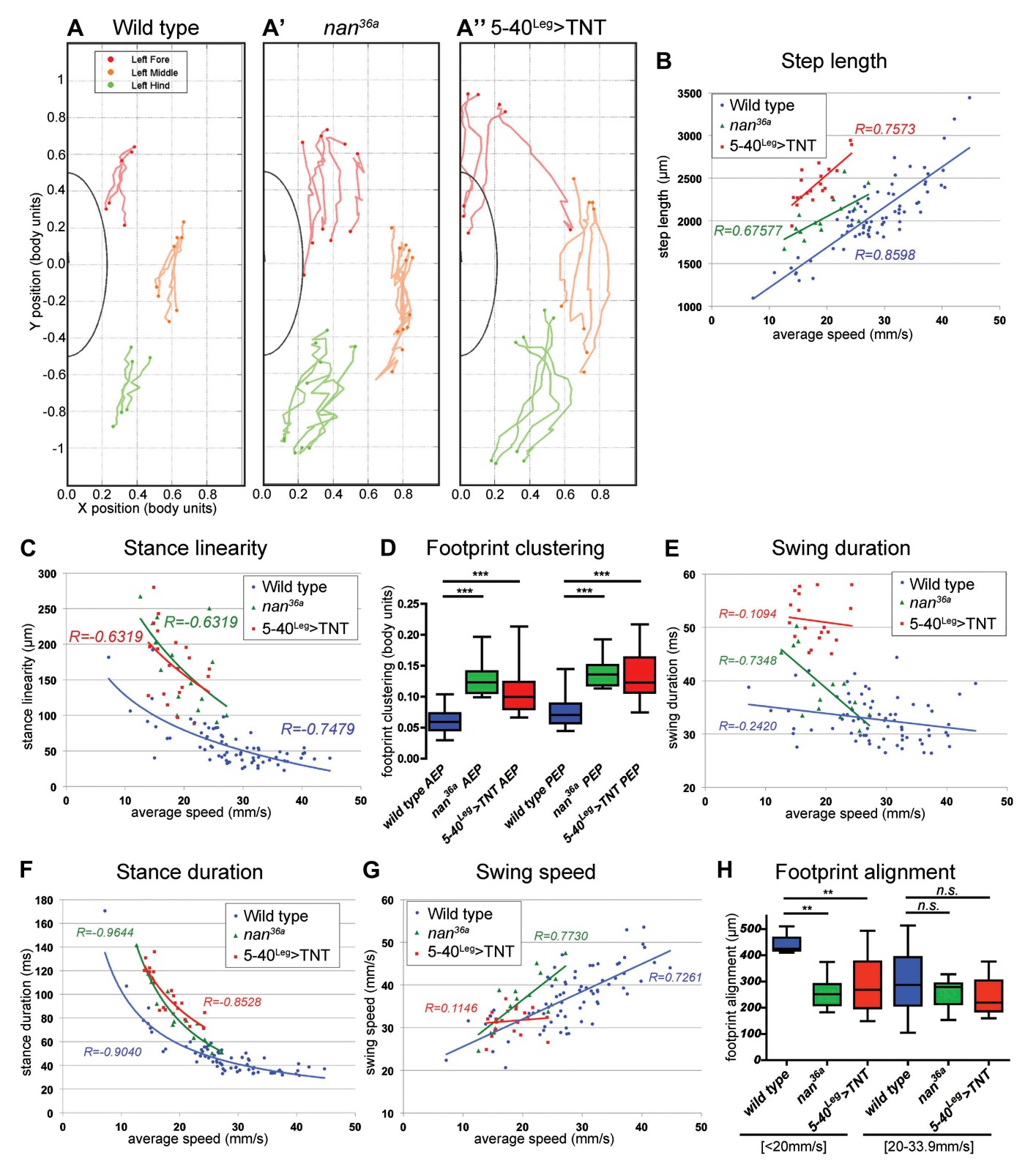

**Figure 7**. Quantification of gait parameters in sensory deprived animals. (**A**). *Stance traces* of three representative animals walking at a similar speed. For simplicity, only left stance traces are shown. Traces for sensory deprived genotypes display a longer step length; higher jitter and a more variable AEP and PEP. (See panels (**B**)–(**D**) and (**H**) for quantification). In (**B**),(**C**) and (**E**)–(**G**), colored lines represent graphical fits. See **Table 2** for statistical analysis of *Figure 7. Continued on next page*

*Figure 7. Continued*

(**B**),(**C**),(**E**),(**F**) and (**G**). (**B**). *Step length*. Sensory deprived animals display an increased step length. (**C**). *Stance linearity*. Sensory deprived animals display a more jittery movement. (**D**). AEP and PEP footprint clustering. Wild type data correspond to the speed range of *nan³⁶ᵃ* and *5-40ᴸᵉᵍ>TNT* (12.6 to 27.7 mm/s). In (**D**),(**H**), boxplots represent the median as the middle line and the lower and upper edges of the boxes representing the 25% and 75% quartiles, respectively; the whiskers represent the range of the full data set. Statistical analysis with Kruskal–Wallis-ANOVA (p<0.0001) followed by Dunn's *post hoc* test, \*\*\*p<0.001. (**E**). Swing duration. (**F**). Stance duration. (**G**). Swing speed. (**H**). *Footprint alignment*. Slow-walking sensory deprived animals display more aligned footprints compared to wild type flies. Data were grouped into slow (<20 mm/s) and medium speeds (between 20 and 34 mm/s). Kruskal–Wallis-ANOVA test: p=0.0003 and NS for the slow and medium speed groups, respectively. Asterisks indicate the significance of the decrease in *footprint clustering* between genotypes. (Data analyzed by the Dunn's *post hoc* significance test, \*\*p<0.005).

by local interneurons to achieve coordination. In contrast, in slower flies, sensory feedback would be invoked to allow animals to negotiate more complex terrains. These speculations are supported by our observations that several parameters measured here, most prominently *footprint clustering*, appear distinct at slow, medium, and fast walking speeds. Thus, although the tripod gait is used at all speeds, distinct neural programs may come into play that are more or less dependent on sensory feedback, depending on the speed.

## Conclusions

The fTIR method and FlyWalker software provides a robust suite of tools for analyzing walking in *Drosophila*. Using this approach, we present many parameters that comprehensively describe walking by wild type flies. Our initial analysis of sensory-deprived flies reveal that proprioception, at least from the legs, is largely dispensable for coordinated walking and the tripod gait. In the future, additional expression tools will allow the targeting of other subsets of sensory and interneurons and the ability to carry out gain-of-function experiments. Moreover, the fTIR apparatus can be used with other arthropods as long as they, like *Drosophila*, possess adhesive structures on their tarsi (*Gorb et al., 2007*). However, although footprints and body position can be readily identified with this method, our current setup does not provide the ability to follow individual leg joints, which can be done in larger insects such as the stick insect (*Cruse and Bartling, 1995*). Nevertheless, the set of parameters measured here underscore the complex mechanisms and circuits regulating hexapod locomotion. Our approach, in combination with the growing collection of genetic tools available in *Drosophila* should open many additional opportunities to unravel the mechanism of locomotion in animals and improve bio-inspired machines.

## Materials and methods

### Fly strains

Oregon R flies were reared on standard cornmeal food at 25°C. *nan-Gal4* (*F-Gal4*) was obtained from the Bloomington Stock Center. *5-40-Gal4* was a gift from Cynthia Hughes (*Hughes and Thomas, 2007*), *UAS-FRT-stop-FRT-TNT* was from Barry Dickson (*Stockinger et al., 2005*) and *nan³⁶ᵃ* was kindly provided by Marco Gallio. *dacᴿᴱ-FLP* was generated by TOPO cloning the Ring Enhancer (*RE*) fragment from the regulatory region of *dac* (*Giorgianni and Mann, 2011*) into an entry clone followed by a Gateway reaction into a FLP destination plasmid. Transgenic lines were generated by standard procedures in a *yw* background. All experiments were carried out with 1 to 7 day old animals at room temperature. To select the correct genotype, flies were anesthetized on a cold plate and allowed to recover at least 24 h. Before imaging, flies were kept in clean glass vials for ~15 min.

### fTIR apparatus

Five Neutral White (4100K) LEDs from Luxeonstar (Brantford, Ontario, Canada), Pre-Mounted on a 10 mm Square Base (230 lm @ 700mA) were wired in series and glued to a 60 mm CPU heat-sink and fan (StarTech.com Lockbourne, Ohio). Three LED/heat-sink sets were wired in parallel and clamped to the edges of a 150 mm × 150 mm × 6.5 mm Borofloat Glass (Advanced Optics, Pewaukee, WI). The edges of the glass were scribed and broken so they would be clear, but not polished. The fourth edge of the glass was fixed to a filter mount and the whole system was

**Table 2.** Multiple regression models for wild type versus sensory-deprived flies[a]

**Wild type vs $nan^{36a}$**

| | Step length (Figure 7B) | | | Stance linearity (Figure 7C) | | | Swing duration (Figure 7E) | | | Swing speed (Figure 7G) | | | Stance duration (Figure 7F) | | | Tripod index (Figure 6D) | | |
|---|---|---|---|---|---|---|---|---|---|---|---|---|---|---|---|---|---|---|
| | Coef | SE | p-value | Coef | SE | p-value | Coef | SE | p-value | Coef | SE | p-value | Coef | SE | p-value | Coef | SE | p-value |
| WT y intercept | 746.7 | 93.8 | 0.0 | 291.6 | 32.6 | 0.0 | $3.6 \times 10^{-2}$ | $1.9 \times 10^{-3}$ | 0.0 | 19.1 | 2.1 | 0.0 | $2.3 \times 10^{-1}$ | $1.2 \times 10^{-2}$ | 0.0 | $-9.2 \times 10^{-4}$ | $4.2 \times 10^{-2}$ | 0.982 |
| Δ y intercept | 587.4 | 270.5 | *0.0* | 349.8 | 96.4 | *0.0* | $2.2 \times 10^{-2}$ | $5.2 \times 10^{-3}$ | *0.0* | −4.5 | 5.9 | **0.451** | $1.9 \times 10^{-1}$ | $3.4 \times 10^{-2}$ | *0.0* | $-4.1 \times 10^{-1}$ | $1.2 \times 10^{-1}$ | *0.001* |
| WT slope | 47.2 | 3.3 | 0.0 | −70.9 | 9.9 | 0.0 | $-1.3 \times 10^{-4}$ | $6.4 \times 10^{-5}$ | 0.043 | 0.6 | 0.1 | 0.0 | $-5.4 \times 10^{-2}$ | $3.5 \times 10^{-3}$ | 0.0 | $1.8 \times 10^{-2}$ | $1.4 \times 10^{-3}$ | 0.0 |
| Δ Slope | −11.3 | 12.9 | **0.4** | −89.1 | 32.1 | *0.007* | $-8.4 \times 10^{-4}$ | $2.5 \times 10^{-4}$ | *0.001* | 0.5 | 0.3 | **0.115** | $-1.6 \times 10^{-2}$ | $4.8 \times 10^{-3}$ | *0.002* | $1.6 \times 10^{-2}$ | $5.6 \times 10^{-3}$ | *0.006* |

**Wild type versus 5-40$^{Leg}$>TNT**

| | Step length (Figure 7B) | | | Stance linearity (Figure 7C) | | | Swing duration (Figure 7E) | | | Swing speed (Figure 7G) | | | Stance duration (Figure 7F) | | | Tripod index (Figure 6D) | | |
|---|---|---|---|---|---|---|---|---|---|---|---|---|---|---|---|---|---|---|
| WT y intercept | 746.7 | 93.8 | 0.0 | 291.6 | 34.0 | 0.0 | $3.6 \times 10^{-2}$ | $1.9 \times 10^{-3}$ | <0.001 | 19.1 | 2.1 | <0.001 | $2.3 \times 10^{-1}$ | $1.2 \times 10^{-2}$ | 0.0 | $-9.2 \times 10^{-4}$ | $4.3 \times 10^{-2}$ | 0.983 |
| Δ y intercept | 622.8 | 295.3 | 0.038 | 242.2 | 115.7 | *0.039* | $1.7 \times 10^{-2}$ | $1.0 \times 10^{-3}$ | *<0.001* | 10.5 | 6.4 | **0.1** | $1.6 \times 10^{-1}$ | $4.1 \times 10^{-2}$ | *0.0* | $-3.6 \times 10^{-1}$ | $1.4 \times 10^{-1}$ | *0.009* |
| WT slope | 47.2 | 3.2 | 0.0 | −70.9 | 10.3 | 0.0 | $-1.0 \times 10^{-4}$ | $6.0 \times 10^{-5}$ | 0.04 | 0.7 | 0.1 | <0.001 | $-5.4 \times 10^{-2}$ | $3.7 \times 10^{-3}$ | 0.0 | $1.8 \times 10^{-2}$ | $1.5 \times 10^{-3}$ | 0.0 |
| Δ Slope | 11.4 | 15.6 | **0.465** | −54.8 | 39.7 | **0.17** | $-2.4 \times 10^{-5}$ | $3.1 \times 10^{-4}$ | **0.94** | −0.5 | 0.3 | **0.12** | $-4.4 \times 10^{-2}$ | $1.4 \times 10^{-2}$ | *0.003* | $1.6 \times 10^{-2}$ | $7.1 \times 10^{-3}$ | *0.032* |

[a]Coef stands for the estimated regression coefficient, SE represents its standard error. In this model we log transformed average speed for parameters that were non-linear with respect to speed (step linearity and stance duration). WT y intercept indicates the y intercept for wild type. Δ y intercept indicates the difference in the y intercept between the experimental condition and wild type. WT slope reports the slopes of the wild type regression lines. p-values >0.05 are in bold italics; those <0.05 are in italics underlined. If Δ slope is >0.05 (bold italics) the regression curves are considered non-interacting (~parallel). If Δ y intercept is <0.05 (italics underlined), the parameter is considered different from WT.

attached to a bench plate by steel mounting posts (Edmund optics, Barrington, NJ). The LEDs were powered by a variable power supply (Tekpower, Montclair, CA). In order to minimize deflected light, the edges of the glass and the LEDs are covered with black ink or black cardboard. Consequently, most of the light from the LEDs enters the glass below the critical angle. To recycle light that reaches the edges of the optical glass, the remaining edges of the glass are covered with aluminum foil. Movies were acquired using a Photron (San Diego, CA) Fastcam MC2 camera using a 55 mm Computar (Commack, NY) telecentric lens, which has very little optical distortion. By measuring 1 mm intervals at multiple positions within the field of view we determined the optical distortion to be less than 4%. The lens aperture (f/2.8) was maximized in order to increase light sensitivity and minimize depth of field.

If the glass–air interface becomes disrupted by the footprints of a fruit fly, the light becomes deflected and is detected by the high-speed video camera. Consequently, when a leg is in stance phase it is detected by fTIR and the absence of signal indicates a swing phase. In addition, a small amount of reflected light illuminates the fly's body (*Figure 1—figure supplement 1*). In the center of the fTIR apparatus, a plexiglass tunnel limits the mobility of the fly and increases the chance it will move in a straight line (*Figure 1—figure supplement 1*). This tunnel is slightly separated from the glass by a nylon string in order to minimize the contact between the plexiglass and optical glass, which would produce a fTIR signal. In order to trigger an optomotor response, two optical fibers connected to a UV light source are present at both ends of the tunnel (*Figure 1—figure supplement 1*). Images are detected within a ~1.2 cm$^2$ area by a high-speed camera at a rate of 250 frames per second at f/2.8. This allows sufficient light to be detected per frame while permitting a high temporal resolution of 4 milliseconds per frame.

For the fly tunnel, a rectangle measuring 7 by 40 mm was cut out of a transparent plexiglass plate measuring 4.76 × 46 × 23 mm. At both ends, a 1 mm diameter hole was made in order to accommodate an optical fiber. As a cover for the tunnel, a second piece plexiglass of equal size was glued on top. A 5 mm hole was drilled on the cover approximately 1 cm from the edge in order to insert the flies into the chamber. In order to raise the chamber slightly, two small holes 1 mm apart, were drilled at each of the four corners of the chamber. A nylon wire—0.48 mm diameter—was inserted through the two holes to create a loop that functions as spacers to prevent the chamber from contacting the optical glass. Finally, the chamber was painted in black on the outside and the interior walls coated with Fluon AD-1 (www.entosupplies.com.au) mixed with black India ink in order to encourage walking on the optical glass (*Figure 1—figure supplement 1*).

## FlyWalker software

FlyWalker was created in MATLAB, and analyzes the sequence of images from the videos by registering the position of the body and each footprint. Once the image sequence is loaded into the FlyWalker interface (*Figure 1—figure supplement 2*), its analysis becomes a two-step process. First, body and footprints are tracked automatically by the software based on sudden increases of brightness within preset thresholds, as well as information from preceding frames. Any fTIR signal present in frames prior to the appearance of the fly (due for example to dust) is subtracted from all subsequent frames to minimize false positive signals. Sequential frame comparison sets the displacement axis, which helps with the identification of each of the six footprints, for example the left and right fore (LF, RF), left and right mid (LM, RM), and left and right hind (LH, RH) legs. Second, the software allows the user to manually correct any mislabeled or missed footprints. A sample video of the processed fTIR effect can be seen in *Video 2*. For each frame, the user can reject a mislabeled footprint or add a footprint that is visible, but was not included because it fell below the preset threshold. The script is optimized to minimize the time spent editing each video. A setup window allows the user to define auto-tracking parameters and which features to show on-screen (*Figure 1—figure supplement 2*). For example, depending on the user-defined settings, the body trace and/or past footprints can be visualized (*Figure 1C*). Importantly, through the use of a calibration reticle, the user can input the pixel/µm ratio, which allows the program to introduce a scale bar (*Figure 1C*), calculate distances and speeds.

## Parameters quantified by FlyWalker

Speed (instantaneous and average)
Frequency
Period

Metachronal lag
Swing speed (average and for individual steps)
Step length (average and for individual steps)
Swing time (average and for individual steps)
Stance time (average and for individual steps)
Footprint alignment (average and for individual footprint clusters)
Anterior Extreme Position (AEP)
Posterior Extreme Position (PEP)
Footprint clustering (AEP and PEP)
Stance trace (average and for individual segments)
Body trace
Tripod index
Tetrapod index
Gait index
Step period (LH:RH; LM:RM; LF:RF; LH:LM; LM:LF; RH:RM; RM:RF)

## List of files generated by FlyWalker

Angle between footprint and displacement axis vs time
Footprint distance to body center vs time
Footprint parallel distance to body center vs time
Footprint perpendicular distance to body center vs time
Instantaneous speed vs time
Gait vs time
Combined gait and instantaneous speed vs time
Anterior extreme position plus stance trace
Posterior extreme position plus stance trace
Geometric combinations generated by footprints
Step size vs time
Gait maps (fixed and automatic time scale)
Gait index vs time (fixed and automatic time scale)
Step velocity; instantaneous speed; color code and gait index plots combined vs time (fixed and automatic time scale).
Summary plots including combined gait and instantaneous speed and anterior extreme position plus stance trace; plus values for average speed, tripod index, tetrapod index, stance trace and average step distance.
Excel file with all parameters
Image sequence from the tracking program.

## Statistical analyses

For most plots, each data point comes from a single video, except for phase and metachronal lag plots where each video generated multiple data points. Because some parameters changed with speed we fit multiple regression models (*Table 2*). The interaction term corresponds to the slope variation (*Δ slope*). When the relationship between the parameter and the average speed was non linear (stance linearity and stance duration), we transformed the average speed by its natural log.

Because of the different behaviors of the slow, medium, and fast speed groups, some parameters were compared between individual speed groups. In these cases, the data were presented as box and whisker plots. For independent observations, comparisons between speed groups (*Figures 3E and 4C*) or between mutant groups (*Figures 6C and 7D,H*) were done using Kruskal–Wallis test followed by Dunn's *post hoc* test (for non-normal distributions) or one-way-ANOVA followed by Tukey's *post hoc* tests (for normal distributions). For paired observations (AEP vs PEP in *Figure 3E* and *Figure 3—figure supplement 3*) with a normal distribution, a paired t-test was used. Otherwise a Wilcoxon signed rank test was used (GraphPad Prism, San Diego, CA).

The least square regression line and the R correlation coefficient are indicated in all scatter plots.

Although *average speeds* of ~20 mm/sec are underrepresented in our collection of wild type videos, the histogram shown in *Figure 2A* is not statistically different from a normal distribution (Shapiro-Wilk test; *p=0.31*). However, a normal probability plot shows non-linearity of the data at

<20 mm/s and greater than ~35 mm/s (data not shown), suggesting that the distribution has significant non-Gaussian outliers in the tails.

## Parameter definitions

Terms and definitions for parameters used in this paper. § indicates definitions described previously by *Strauss and Heisenberg 1990* (and references within).

### Period§

Time taken to complete one leg cycle consisting of one swing and one stance phase.

### Step length§

Distance between two successive footprints of the same leg. No normalization in relation of the direction of propagation.

### Stance linearity index

Average difference between the stance traces generated by each leg during stance phase and a 5-point smoothed line.

### Anterior Extreme Position (AEP) (*Cruse, 1976*)

Position where the leg first contacts the glass after touchdown at the end of swing phase (or protraction).

### Posterior Extreme Position (PEP) (*Cruse, 1976*)

Position at the end of the stance phase, just before the tarsi enter swing phase.

### Footprint clustering

Standard deviation from the average position for all *AEP*s or *PEP*s.

### $_X Lag_Y$§

Time interval between the beginning of swing phase in leg x and immediate following swing phase onset of another leg y.

### Metachronal Lag§

Time interval between the beginning of swing phase in hindleg and immediate following swing phase onset of the ipsilateral foreleg ($_H Lag_F$).

### Phase§

Lag between two legs divided by the period of the first.

### Tripod index

Percentage of frames in a video that display leg combinations defined by the tripod gait.

### Tetrapod index

Percentage of frames in a video that display leg combinations defined by the tetrapod gait.

### Gait index

Average value for a window of frames where tripod configurations = +1 value; tetrapod = −1; and noncanonical = 0.

### Footprint alignment

Standard deviation from the average point of adjacent ipsilateral footprints projected onto the displacement axis. Each set of ipsilateral footprints includes one foreleg, one midleg and one hindleg.

## Acknowledgements

We thank the Bloomington Stock Center, Barry Dickson, Marco Gallio, Cynthia Hughes and Matt Giorgianni for reagents. We also thank Ioannis Kymissis for initial advice on this project, Charles Zuker and members of the Mann laboratory for encouragement, suggestions and comments on the manuscript. We also thank the Irving Institute (http://irvinginstitute.columbia.edu/resources/biostatistics.html; Grant Number UL1 RR024156) and in particular Jimmy Duong and Arthur Palmer for assistance with the statistical analysis.

## Additional information

### Funding

| Funder | Grant reference number | Author |
|---|---|---|
| National Institutes of Health | R01NS070644 | César S Mendes, Richard S Mann |
| Project ALS | | César S Mendes, Richard S Mann |
| Motor Neuron Center, Columbia University | | César S Mendes, Turgay Akay, Richard S Mann |
| Fundação para a Ciência e a Tecnologia, Portugal | | César S Mendes |
| Ellison Medical Foundation | | César S Mendes, Richard S Mann |

The funders had no role in study design, data collection and interpretation, or the decision to submit the work for publication.

### Author contributions

CSM, Conception and design, Acquisition of data, Analysis and interpretation of data, Drafting or revising the article; IB, Conception and design, Analysis and interpretation of data, Drafting or revising the article; TA, Conception and design, Analysis and interpretation of data, Drafting or revising the article; SM, Conception and design, Drafting or revising the article; RSM, Conception and design, Analysis and interpretation of data, Drafting or revising the article

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
