## [Author Response]

*1. While the writing is generally clear, there is concern that the Introduction presents a naïve view of how CPGs work and interact with descending inputs and sensory feedback. A primary interest of the manuscript is what it tells us about how sensory inputs interact with the locomotory CPG. Specifically, the opening paragraph of the manuscript should be reworked to reflect the more sophisticated view of CPGs put forth by the references cited so that this interest is properly showcased*.

We have revised the first and second paragraphs accordingly.

*2. While this manuscript was under review, another manuscript concerning fly walking has been accepted for publication. Wosnitza A, Bockemühl T, Dübbert M, Scholz H, Büschges A. Inter-leg coordination in the control of walking speed in Drosophila. J Exp Biol. 2012 Oct 4. [Epub ahead of print] PubMed PMID: 23038731. This paper should be cited and the results contrasted; in particular a contrast should be made between leg amputation and sensory inactivation as a means of exploring how sensory inputs interact with the locomotory CPGs. In that manuscript all speed data are shown in body lengths so comparisons will be difficult but should nevertheless be made*.

We now cite Wosnitza et al several times in the manuscript (in the Introduction and Discussion) and mention how their leg amputation data fit with our findings.

*3. There were significant concerns about some of the data analysis*.

*a) The authors should more explicitly rationalize splitting the distribution of Figure 2A into three speed groups. (An external reviewer wrote “Although the data showing differences between fast and slow animals are fairly convincing I'd like to see a rationale for dividing the animals in 3 groups (why not 2 groups: above and below average; or 4 groups?). The way they group animals is going to affect their statistical analysis.”)*.

There are several independent results in our paper that support the division of the data into three speed groups. Perhaps the most clear of these are the data presented in Figure 5B (footprint alignment vs speed), where the data naturally fall into three distinct groups (<20 mm/sec, >20 to <34 mm/sec, and >34 mm/sec). Second, there is a dip in the number of flies walking at average speed values of ∼20 mm/sec (Figure 2A). Third, the stance linearity data shown in Figure 3C argue that speeds >34 mm/sec behave differently than speeds < 34mm/sec. Fourth, normal probability plots are linear between ∼20 and ∼35 mm/sec, but show non-linearity above and below this range (see our response to comment 3d, below, to see this plot). Thus, the two speeds (∼20 and ∼34 mm/sec) that define our three speed groups were chosen due to non-linear or discontinuous behaviors of the data derived from four independent readouts.

Although the data supporting these three speed groups are in separate figures, we now explicitly describe this early in the paper when it is first raised, referring briefly to the later figures.

*b) The distribution of Figure 2A does suggest a break at a speed of 20mm/s. Is there a statistical test that can verify this split in the distribution or can more animals at low speed be measured so that a clear break can be seen*?

We carried out multiple statistical tests to determine if the speed distributions shown in Figure 2A fit a normal distribution. In all cases, the tests (Kolmogorov-Smirnov, D’Agostino and Pearson, and Shapiro–Wilk) failed to support a significant deviation from a normal distribution, which is what we reported in the paper. However, as detailed below (in response to comment d), a normal probability plot revealed that the tails of the distribution have significant outliers. Notably, deviations from linearity in these plots occur at ∼20 and ∼34 mm/sec (see plot below).

Because of the nature of the speed distribution data, it is very difficult to obtain additional videos of flies walking at slow speeds – approximately 8 of 10 videos show average speeds >20 mm/sec – making it impractical to increase the numbers sufficiently to improve the statistical analysis.

In summary, although the full speed data set fits the strict definition of a normal distribution, speeds at ∼20 mm/sec are nevertheless underrepresented. A normal probability plot reveals that the data below and above 20 and 34 mm/sec, respectively, are not well modeled by a normal distribution. All of this is now better explained in the paper.

*c) In the Materials and methods the authors state: “Because of the different behaviors of the slow, medium, and fast speed groups, some parameters were compared between individual speed groups. In these cases, the data were presented as box and whisker plots. When the data sets had a normal distribution, a Student's t-test was used. Otherwise a Mann-Whitney U-test was used.” There was general concern that these statistical methods were not adequate and that some form of ANOVA (or equivalent non-parametric test), followed by appropriate post-hoc tests, should be pursued*.

We revised our statistical analysis when comparing measurements between the three speed groups. For group comparisons that seem to have normal distributions, we used one-way-ANOVA followed by Tukey’s post hoc test for pairwise comparisons. For group comparisons where the distributions seemed non-normal, a Kruskal–Wallis test followed by Dunn’s post hoc tests for pairwise comparisons were used. Moreover, comparisons between AEP and PEP (within the same speed group) were revised taking into account that these two parameters are paired (i.e., the same video generated one AEP and PEP value). Depending on whether the groups had normal or non-normal distributions, we used the paired t-test or Wilcoxon signed rank test, respectively.

Accordingly, we updated the *Statistical analysis* section in the *Materials and methods*, the figures, and figure legends. Importantly, these changes do not alter our conclusions.

*d) The legend to Figure 2A states: “Speed histogram of 71 videos recorded for wild type flies, with 2 mm/s bins. Average speeds vary between 7.2 and 44.7 mm/s, with 28 mm/s the most represented speed. Although the data in this histogram fit a normal distribution, speeds of 20 mm/s are underrepresented. Additionally, both tails of the distribution curve have significant outliers (data not shown).” How were outliers determined and on what basis were they then excluded*?

No data were excluded from this analysis.

The statement that “both tails of the distribution curve have significant outliers” was based on a normal probability plot. For a normal distribution, these plots should be linear. However, for the speed data shown in Figure 2A, the data are linear for speeds between 20 and ∼34 mm/sec, but then non-linear for speeds <20 mm/sec and >∼34 mm/sec. Interestingly, the transitions from linearity match very closely to the divisions between the three speed groups as defined above, providing additional support for these three speed groups. The normal distribution plot is shown below.

For simplicity, we have clarified the above points in the text, and refer to the normal probability plot as ‘data not shown’ in the *Statistical analysis* section in the *Materials and methods*.